# Manganese-Induced Parkinsonism: Evidence from Epidemiological and Experimental Studies

**DOI:** 10.3390/biom13081190

**Published:** 2023-07-30

**Authors:** Roberto Lucchini, Kim Tieu

**Affiliations:** 1Department of Environmental Health Sciences, Florida International University, Miami, FL 33199, USA; 2Biomolecular Sciences Institute, Florida International University, Miami, FL 33199, USA

**Keywords:** manganese, manganism, neurotoxicity, parkinsonism, Parkinson’s disease, gene-environment interaction, α-synuclein, parkin, DJ-1, ATP13A2

## Abstract

Manganese (Mn) exposure has evolved from acute, high-level exposure causing manganism to low, chronic lifetime exposure. In this latter scenario, the target areas extend beyond the globus pallidus (as seen with manganism) to the entire basal ganglia, including the substantia nigra pars compacta. This change of exposure paradigm has prompted numerous epidemiological investigations of the occurrence of Parkinson’s disease (PD), or parkinsonism, due to the long-term impact of Mn. In parallel, experimental research has focused on the underlying pathogenic mechanisms of Mn and its interactions with genetic susceptibility. In this review, we provide evidence from both types of studies, with the aim to link the epidemiological data with the potential mechanistic interpretation.

## 1. Introduction

Two decades after James Parkinson’s Essay on The Shaking Palsy was published in 1817, John Couper published his landmark article, On the Effects of Black Oxide of Manganese when Inhaled into the Lungs [1]. Since then, ‘manganism’ and parkinsonism have crossed paths, creating the uncertainty regarding their differences and similarities. Throughout the years, technological changes have increased human exposure to Mn, mirroring now more closely the phenotypes of parkinsonism [2]. Today, the industrial production and utilization of Mn have sharply increased, mainly due to the high demand for steel required for construction in emerging economies such as China, India, and the Middle East. The discovery of better-performance Mn-based batteries for electric vehicles is projected to further increase the production of pure Mn and its global utilization, replacing the more expensive cobalt-based batteries. Welding will also keep growing, with consequent exposure of welders to much smaller Mn-containing particles. All this is reflected by the progressive increases in metal pollution in soil [3,4,5,6] related to anthropogenic activities. Few data are available on the presence of Mn in the ocean and aquifers [7,8] but available data have suggested areas with high levels in drinking water. Air concentration is highly variable, depending on the presence of industrial emissions. Car traffic does not seem to be a source of pollution, as the Mn concentration in gasoline, mainly in the form of MMT, seems to be negligeable.

A full international conference on Mn toxicity took place in 2016 at the Icahn Scholl of Medicine at Mount Sinai, New York. New epidemiological, toxicological and cellular studies reported at the conference yielded new insights into mechanisms of Mn toxicity and opportunities for preventive intervention. Strong evidence was provided for causal associations between Mn and both neurodevelopmental and neurodegenerative disorders. Brain imaging data strongly substantiated the new findings at different life stages. Candidate biomarkers of exposure were present for hair, nails, and teeth, reflecting different exposure windows and outcomes [9]. Sex differences have been reported in several studies, suggesting that women are more susceptible, due to less efficiency in managing Mn transport, which is regulated by the SLC30A10 and SLC39A8 transporters, resulting in disturbed Mn homeostasis and toxicity [9].

Whether Mn exposure is associated with Parkinson’s disease (PD) or parkinsonism, and their differences and similarities with manganism, the clinical intoxication resulting from acute high exposure to Mn, have been topics of debate. Globally, the ‘Parkinson’s pandemic’ is the fastest growing neurological disease and has increased from 2.6 million in 1990 to 6.3 million in 2015. By 2040, it is projected to increase to 17.5 million not only due to aging but also to increasing longevity and industrialization [10]. In North America, the most updated information on PD frequency, based on multi-study and Medicare sources indicates that it may be higher than what has previously been reported. The prevalence rate based on data collected in 2010 among those aged ≥45 years was 572 per 100,000 (95% confidence interval 537–614), with 680,000 patients in the US aged ≥45 years. Based on the US Census Bureau’s population projections, that number will double in 2030 [11]. Incidence rates calculated from the same sources in 2012 were 47–77/100,00 among those aged ≥45, and 108–212/100,000 among those aged ≥65 [12]. Age- and sex-adjusted analyses also show spatial clustering in southern California, southeastern Texas, central Pennsylvania, and Florida, possibly due to case ascertainment and diagnostic protocols, and environmental exposure in certain geographic location.

A growing interest has developed in the past two decades in the possibility that Mn exposure acts as an exogenous trigger to induce parkinsonism, which is different from the classical form of manganism. Human epidemiological studies indicate that increased frequency of parkinsonism is associated with chronic Mn exposure, which, although not reaching sufficiently high levels to cause acute manganism, is sufficient to induce parkinsonian symptoms [13,14,15,16]. Clinical cases of parkinsonism have been described in relation to the homozygous mutations in the solute carrier (SLC) transporters, which mediate the influx (SLC39A8, SLC39A14) and efflux (SLC30A10) of Mn [17,18,19,20,21]. Human observations have been supported also by experimental work, which provides plausible mechanistic explanations. The clinical feature of Mn-induced parkinsonism is now considered as a different phenotype from idiopathic Parkinson’s Disease (IPD), based on the presence of four cardinal signs of rest tremor, bradykinesia, rigidity, and impaired postural reflexes. Mn-induced parkinsonism includes a broader classification defined by the presence of at least two of the four cardinal signs [22] and the benchmark dose for Mn concentration in PM10 airborne particles has been estimated in 20–25 ng/m^3^, as a cut-off for increased risk [23]. Classical manganism is instead the result of acute exposure to high Mn concentrations of at least 1 mg/m^3^, as indicated by WHO [24].

Accurate descriptions of the clinical manifestation of that atypical parkinsonism have been provided by several reports starting from Dr. Couper’s paper [1], to similar reports from Morrocco [25], Chile [26,27,28]. They all indicate slowness of movement (bradykinesia), masked facies, and gait impairment (postural instability) as the main features, in addition to the lack of response to L-Dopa treatment. Cases of clinical manganism have been reported more recently in China, where ferroalloy production is highly developed and not always sufficiently controlled by modern preventive intervention [29,30]. Not only inhalation can be the cause of Mn overload, causing manganism, but also intravenous absorption of excessive Mn through parenteral nutrition, calling for a more adequate regulation of Mn concentrations in parenteral solution [31,32,33]. Consumption of illicit drugs such as ephedrone, obtained via the reaction of pseudoephedrine with potassium permanganate (KMnO_4_), acetic acid, and water. The oxidation of ephedrine/pseudoephedrine-induced KMnO_4_ forms methcathinone (ephedrone) and manganese dioxide, resulting in clinical manganism in young adults [34,35]. Finally, Mn overload leading to clinical manganism is caused by hepatic portosystemic shunts in liver failure conditions, blocking Mn excretion through the biliaric system and resulting in acquired hepatocerebral degeneration [36].

Treatment of these clinical manifestations has shown somewhat inconsistent results after para-aminosalicylic acid (PAS) [37,38], ethylenediaminetetraacetic acid (EDTA) [39,40], and 2,3 dimercaptosuccinic acid chelation [41]. Removing Mn overload is currently the only treatment of genetically induced manganism due to homozygous mutation of Mn transporters and is able to reduce the severity of symptoms, to an extent, among affected children.

These data indicate that Mn exposure poses a high public health concern for the future, and together with the increasingly diffused presence of Mn in workplaces and the environment, call for a clear definition of the role of Mn. With this review, we provide further information and updates, focusing on Mn as a potential risk factor for parkinsonian outcomes based on human epidemiological studies and experimental mechanistic studies.

## 2. Methods

We conducted a literature search using the keywords, “manganese”, “parkinsonism”, “Parkinson’s disease”. We utilized the following databases: Scopus, MedLine, and PubMed. The databases were chosen due to their great variety and the broad scope of available studies. The search was conducted based on single MeSH terms or in combination. Each manuscript was reviewed by RL and KT and classified per type of design.

## 3. Human Studies

A variety of studies have been conducted on human populations exposed to Mn, through their occupation or environmental exposure, to fill the gaps in knowledge regarding Mn as a potential risk factor for PD and/or parkinsonism. This review focuses on human studies published after a previous review in 2009 by Lucchini et al. [13]. That analysis concluded that although acute manganism is a distinct medical condition from PD, the progressive changes of exposure scenarios towards chronic exposure to much lower levels (see Figure 1) might have progressively extended the site of Mn deposition and toxicity from the globus pallidus to the entire area of the basal ganglia, including the substantia nigra pars compacta, a brain region affected in PD.

### 3.1. Occupational Studies

Welders have been the subject of human studies exposed to Mn through their occupation for many years, starting from the first publication by Racette et al. in 2001 [14], reporting an early onset of PD among 15 career welders who showed a reduction in dopamine through LEVODOPA PET scans and were responsive to dopamine treatment. This research group has produced a few subsequent papers as discussed below, reinforcing the finding of high PD incidence among welders, with no substantial differences from non-welder PD patients.

In 2012, Racette et al. examined 811 shipyard and construction welders and compared them to 59 non-welder workers and 118 patients diagnosed with idiopathic PD [15]. The Unified Parkinson’s Disease Rating Scale, motor sub-section 3 (UPDRS-3), was utilized for the diagnostic assessment. UPRDS-3 scores higher than 15 were considered parkinsonism and were found in 15.6% of the welders compared to 0% in the reference group. Welders showed a U-shaped dose–response relationship between exposure years and parkinsonism. UPDRS-3 scores among welders were like the newly diagnosed idiopathic PD cases that showed more frequent resting tremors and asymmetry.

In 2015, Andruska and Racette reported a prevalence of parkinsonism of 15.6% among 716 welders and a PD prevalence of 2% in the general population over 65 years of age [16]. The studies by Racette’s group have consistently showed a relationship between Mn exposure and parkinsonism, compared to other negative studies based on clinical record or metanalysis [42,43,44,45] that did not rely on the same methodology based on individual assessment using UPDRS-3.

Modern FDOPA PET research in pre-symptomatic Mn welders has also shown impaired FDOPA uptake in the striatum and lower uptake in the caudate compared to the putamen compared to non-exposed subjects [46,47]. This is in contrast with previous studies not showing altered FDOPA uptake in Mn exposed or intoxicated subjects. The improvement of FDOPA PET technology and the changes in the exposure patterns may explain this difference in imaging findings.

Mn is always present in welding fumes and therefore, it is biologically feasible that at the current average exposure levels in welding, Mn toxicity may be insufficient to produce clinical manganism but sufficient to enhance the effects of a reduced release of dopamine, resulting in presynaptic dysfunction and PD at an earlier stage [48]. Low-level Mn exposure over time may more accurately mimic environmental triggers that have been associated with PD risk [16].

Among 418 Mn miners in South Africa, parkinsonism defined by a UPDRS-3 score higher than 15 showed a prevalence of 29.4% [49]. For Iranian automotive workers, the rate of parkinsonism assessed with transcranial sonography was 42% [50]. These workers had been exposed to 3.34 mg/m^3^ of cumulative Mn on average, over an average duration of 12.30 years of work, whereas the South African miners had experienced a similar average cumulative exposure of 3.7 mg/m^3^ over an average duration of 13.5 years.

### 3.2. Environmental Studies

Non-occupational studies have been conducted to assess the relationship between Mn exposure and PD and/or parkinsonism. They have mainly included ecological and case–control designs.

A high prevalence of parkinsonism was observed in the province of Brescia, Italy, where clustering of Bayesian SMRs (BSMRs) resulted in the vicinities of ferroalloys industries operating since the beginning of 1900. A significant correlation was observed between the BSMRs for parkinsonism and the average concentration of Mn in the deposited dust [51]. In the same province, case–control studies have shown a relationship of PD and parkinsonism with metal exposure and α-synuclein polymorphism [52]. Being born in the province of Brescia also resulted in a potential determinant of early life exposure in the development of the disease at the old age [52]. The parkinsonian patients in this area showed a serum increase in liver enzymes AST/ALT, together with higher levels of blood and urinary Mn, and disruption of copper, zinc, and iron levels [53]. The involvement of liver function in patients exposed to Mn indicates a potential link to genetic mutation of the SLC enzymes that regulate Mn transportation, as seen in other studies [54].

Homozygous mutations of this family of transporters (*SLC30A10*, *SCL39A8*, *SLC39A14*) cause a rare pediatric genetic disease characterized by hypermagnesemia, parkinsonism and dystonia, liver disease and polycythemia [55,56,57,58], and specific imaging signs [59]. Heterozygous mutations of these transporters are commonly seen in up to 40% of the cases in the province of Brescia [60]; they influence Mn in blood, leading to neurological impacts [61,62]. Therefore, a Mn-induced increase in PD and parkinsonism in the Italian site may be related to historic environmental contamination and genetic vulnerability. The province of Brescia is a unique site impacted for more than a century by ferromanganese emission in enclosed pre-Alp valleys like Valcamonica, resulting in high levels of Mn in soil [63,64,65], local products [66,67], and deposited dust [68,69], causing continuous airborne resuspension [70,71]. Epidemiological studies focused on these types of locations cannot be compared to others where no such contamination is present.

### 3.3. Diagnostic Criteria

According to the evolving exposure scenario, diagnostic assessment is also changing. While the response to levodopa has been used in the differential diagnosis of manganism from IPD, this diagnostic feature no longer reflects the different toxicological target. From the manganism-localized toxicity delimited to the globus pallidus, Mn-induced parkinsonism reflects toxicity in a larger area of the basal ganglia, including the substantia nigra pars compacta. Therefore, individuals affected by parkinsonism after long-term exposure to low Mn levels can respond to levodopa [72]. The globus pallidus hyperintensity, considered as an imaging hallmark of manganism, may not be detected any longer, after chronic exposure to low Mn levels. Depending upon the individual exposure intensity, duration, timing, co-exposure with other neurotoxicants, genetic predisposition given by the diffuse non-homozigous mutation of the SLC transporters, the clinical diagnosis may reveal an overlap in syndromes between manganism, Mn-induced parkinsonism, and IPD (see the mixed diagnosis overlapping area in Figure 1. This is a more challenging situation compared to the past; therefore, diagnosis requires, in the first place, knowledge and clinical suspicion. The assessment of modern Mn neurotoxicity requires a multidisciplinary approach, linking early recognition to outpatient referral to neurology for definitive care. Usage of MRI or serum-based studies should be done at the request of specialists familiar with toxicity and the latest research [73]. Increased levels of misfolded α-synuclein in central-nervous-system-derived exosomes can be observed both in Mn exposed individuals [74] and parkinsonian patients [75]. Therefore, although the diagnostic predictivity of Mn-induced parkinsonism is still not established, the increased production of toxic α-synuclein in exosomes can provide the mechanistic foundation of Mn-induced parkinsonism. Notably, misfolded α-synuclein is associated with an earlier onset of parkinsonism, which has been observed among Mn-exposed individuals [15,76,77].

## 4. Experimental Studies

As discussed above in the human studies, manganism and PD are two distinctive disorders [78,79,80]. However, Mn may be a risk factor for developing PD by interacting with an individual’s genetic makeup [23,81]. Such interactions have been investigated in mammalian cell culture and animal studies where Mn was combined with PD-linked genes such as *SNCA*, *parkin*, *DJ-1*, and *ATP13A2* [82,83,84]. Given that such gene products and Mn share pathogenic mechanisms such as mitochondrial impairment, neuroinflammation, oxidative stress, and protein misfolding [82], it is not surprising that when combined, their neurotoxic effects are amplified. Below are some genes that have been documented to interact with Mn.

### 4.1. SNCA

*SNCA* encodes α-synuclein, which is predominantly a synaptic protein that is encoded by the *SNCA* gene. Missense mutations as well as mutations leading to duplications and triplications of *SNCA* have been identified in autosomal dominant PD [85,86,87,88,89,90]. The discoveries that increasing the gene dosage of *SNCA* by 2- to 3-fold can cause PD [89] is significant for idiopathic PD, because it indicates that elevated wild-type (WT) α-synuclein alone is sufficient to cause the disease. In addition to familial PD, α-synuclein aggregation is also detected in the Lewy bodies of idiopathic PD [91]. Lewy bodies are intracellular inclusions that consist of aggregated α-synuclein, and more recently, they have also been reported to contain lipids and organelles such as mitochondria [92]. Pathogenic mechanisms associated with α-synuclein mutations range from impairing protein degradation pathways (autophagy and ubiquitin proteasomal system), mitochondrial dysfunction, oxidative stress, synaptic dysfunction, and neuroinflammation [93]. There has also been a significant interest in the spread of α-synuclein pathology from one cell to another in a prion-like fashion [75] and Mn has been demonstrated to enhance this transmission [74].

Mn can bind to α-synuclein via three residues in the C-terminal domain: Asp-121, Asn-122, and Glu-123 [94]. Although this is a low-affinity binding, low concentrations of Mn are sufficient to induce α-synuclein fibril formation [95]. Several independent laboratories have reported the effects of Mn on the aggregation of α-synuclein [96]. The interaction between α-synuclein and Mn has been demonstrated in rat primary midbrain dopaminergic neurons, resulting in a higher intracellular level of Mn in cells with α-synuclein overexpression [97,98]. Although such binding of α-synuclein is protective against Mn neurotoxicity during the early stages of Mn exposure, the accumulation of sequestered Mn by α-synuclein eventually induces protein aggregation and neurotoxicity [99]. In a more recent study, these investigators demonstrated that when combined with α-synuclein, Mn exposure increases α-synuclein transmission intercellularly via exosome release [74]. By spreading the misfolded toxic α-synuclein, these exosomes induce neuroinflammation and degeneration of the nigral dopaminergic neurons [74]. This is the first in vivo demonstration that, although Mn by itself does not cause neurodegeneration, when combined with a PD-linked protein, nigral dopaminergic cell loss occurs. The results of these studies are consistent with a previous report demonstrating that α-synuclein overexpressing neuronal cells released exosomes capable of transferring α-syn protein to other normal neuronal cells [100], forming aggregates and inducing death in the receiving cell [101,102]. In fact, exosome-associated α-synuclein oligomers are more likely to be taken up by recipient cells and are more neurotoxic compared to free α-syn oligomers [103]. To enhance the relevance of their mouse study, Kanthasamy and colleagues isolated exosomes in the serum of welders exposed to Mn and they found increased misfolded α-synuclein in these workers as compared to the non-exposed control subjects [74]. In non-human primates, Mn exposure has also been reported to induce α-synuclein aggregation [104]. In combination, emerging evidence indicates that Mn promotes α-synuclein aggregation and spread, leading to dopaminergic neurodegeneration in experimental models and suggests that such a scenario may also occur in humans [27].

### 4.2. Parkin

Mutations in *Parkin* cause autosomal recessive early-onset PD [105,106], which may present with or without Lewy bodies [106,107]. *Parkin* mutations are considered as the most prevalent autosomal recessive mutations in PD, accounting for approximately up to 77% of familial early-onset PD and 10–20% of early-onset PD in general [108,109,110]. Parkin is an E3 ubiquitin ligase that mediates protein degradation. Parkin has been reported to regulate Mn transport via the divalent metal transporter 1 (DMT1) [111]. DMT1 is the primary route by which Mn is taken up in the brain. Four DMT1 isoforms have been identified and the 1B isoform is regulated post-translationally and degraded via the proteasomal pathway. Parkin is involved in the ubiquitination and mediate the subsequent proteasomal degradation of this specific isoform of DMT1 [111]. Therefore, a loss of parkin function could facilitate Mn uptake, promoting Mn accumulation in the basal ganglia and accelerating its toxicity.

### 4.3. DJ-1

Mutations in *DJ-1* cause early-onset autosomal recessive PD [112]. DJ-1 is a multifunctional protein. Under basal conditions, DJ-1 is localized mostly in the cytoplasm and to a lesser extent, in the mitochondria and nucleus [113,114,115]. However, under oxidative stress conditions, cytoplasmic DJ-1 translocates to mitochondria [115], where it confers protection against oxidative stress [115,116]. Mn has been reported to reduce the levels of DJ-1, resulting in effects that resemble the neurotoxic activity of mutant forms of DJ-1 [117]. Given that both Mn and loss of DJ-1 function can individually promote oxidative stress and induce mitochondrial dysfunction, such combination may exacerbate their neurotoxic effects. Indeed, the life-span of *C. elegans* was reduced after Mn exposure when DJ-1 was deleted [118].

### 4.4. ATP13A2

In addition to causing autosomal recessive juvenile-onset Kufor–Rakeb syndrome [119], which is a levodopa-responsive form of pallido-pyramidal neurodegeration, homozygous missense mutations in *ATP13A2* cause juvenile parkinsonism and early-onset autosomal recessive PD [120]. *ATP13A2* encodes ATPase type 13A, a lysosomal P5B-type transmembrane ATPase which has functional domains like other P-type ATPases that are mainly involved in transporting cations, including Mn. Consistent with these roles of ATP13A2 and its cellular localization, loss of function in this protein has been reported to impair autophagy flux and accumulation of aggregated α-synuclein [121,122,123]. ATP13A2 has been shown to protect cells against Mn toxicity by transporting Mn into lysosomes; however, this protection is lost when mutations in this gene occur [123]. A recent study has shown that ATP13A2 polymorphism could negatively impact the effects of Mn on motor coordination in humans exposed to Mn [124].

## 5. Discussion

This review indicates that Mn-induced parkinsonism should be given a current updated and structured nosological classification that is clearly different from the historical knowledge of classic manganism. The evolution of the exposure scenario drives this epidemiological and clinical redefinition of Mn’s impact on the central nervous system. Manganism is still occurring, resulting from any situation causing Mn overload exceeding the Mn homeostatic range in an acute manner. No matter what the cause is, from high Mn levels in parenteral nutrition, or the consumption of illicit drugs like ephedone, hepatic failure blocking biliaric excretion, genetic dysfunction of Mn transporters, or a lack of exposure control in the workplace resulting in airborne Mn concentrations higher than the WHO cut-off of 1 mg/m^3^, the features of this atypical parkinsonism, known as manganism since 1837, can still affect a substantial number of patients. L-DOPA and chelation with EDTA and PAS are used to reduce the symptoms with varying results due to individual factors that can influence the therapeutic response.

Mn-induced parkinsonism is likely more diffuse in the population, mainly because of ambient levels that easily exceed the cut off values of 20 ng/m^3^ that have resulted from academic research and the reference concentration of 50 ng/m^3^ resulting from the risk assessment estimates by US EPA and Health Canada. Therefore, given the relevance of the constantly evolving exposure scenario, further research is needed to target detailed exposure assessment of the emerging occupational and environmental sources including (i) steel production for growing demand in construction, and (ii) production, use and especially disposal of Mn-based EV batteries. Intervention studies focused on the effectiveness of exposure reduction are also envisaged, given the demonstrated potential for long-term neurological impacts of low doses. Mixture studies are especially needed to quantify the weight of Mn when co-exposures with neurotoxic and protective elements occur. In this regard, novel models such as Bayesian kernel machine regression [125] and weighted quantile sum regression [126], enable understanding of the interaction of Mn with other protective and toxic elements and assessment of both the mixture’s effect and the role of each component within the mixture.

Experimental models used to study neurotoxic mechanisms and the neurotoxicity of Mn should take into consideration the Mn dosage regimen. Acute high doses are more relevant to manganism whereas a low chronic dose is more appropriate for parkinsonism and PD. Different concentrations of Mn may also differentially impair cellular mechanisms. For example, although Mn is well-documented to block mitochondrial function, a recent study has demonstrated that when exposed to concentrations below the acute cytotoxic threshold, Mn does not induce mitochondrial dysfunction in neuronal cells [127]. Another important consideration is the use of animal models when feasible. Although in vitro models have their role, given the complexity and the intricate interactions between cell types in the brain, in vivo models will allow investigations of Mn-induced toxicity in specific cell types in different brain regions.

## 6. Conclusions

Mn-induced manganism is a well-accepted neurotoxic impact of Mn. What is less clear, however, is whether Mn plays a pathogenic role in parkinsonism and, especially, in idiopathic PD. There are overlapping clinical and neuropathological features between these neurological conditions, suggesting some common brain regions and pathogenic mechanisms being involved. The basal ganglia are the region commonly affected between Mn- and PD-linked proteins. Additionally, there are common mechanisms such as mitochondrial dysfunction, neuroinflammation, oxidative stress, and dysregulated protein homeostasis [82,128]. We believe, as illustrated in Figure 1, that a combination of the duration of Mn exposure, exposure intensity, and genetic susceptibility influences the outcome of Mn-induced neurotoxicity. Acute high dose exposure typically causes manganism and the globus pallidus is the primary target. Chronic, low-level exposure extends Mn deposition and toxicity to other brain regions, including the substantia nigra [129]. When combined with other PD-linked gene products such as α-synuclein, parkin, DJ-1, and ATP13A2, it is likely that Mn contributes to the onset and progression of idiopathic PD. In addition to the ones discussed in this review, Mn would most likely also interact and enhance the parkinsonian effects of other genes. Cumulatively, based on the human and experimental studies, as exemplified in this review, Mn most likely contributes to parkinsonism/PD pathogenesis; therefore, it is imperative to control this modifiable factor and eliminate all possible causes of overexposure and long-term absorption of low doses. This call for preventive intervention was clearly stated by Dr Couper almost two hundred years ago [1] and must be further embraced today in view of the projected exponential surge of PD over the next two decades [10].

## Figures and Tables

**Figure 1 biomolecules-13-01190-f001:**
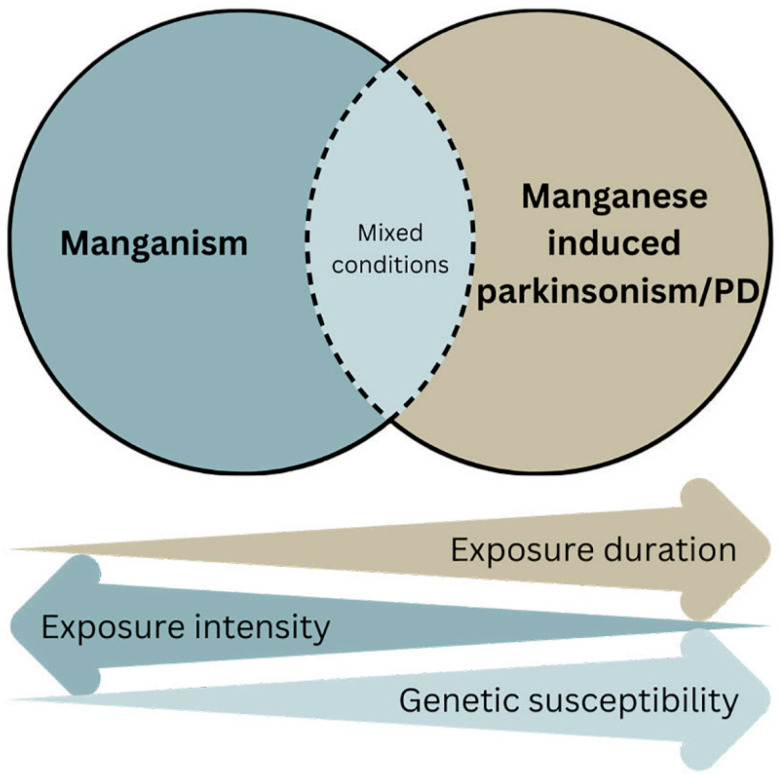
Occurrence of manganism, manganese-induced parkinsonism and/or PD, and intermediate mixed conditions as a function of exposure intensity, exposure duration and genetic susceptibility.

## Data Availability

Not applicable.

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
