# Peer review of "Manganese-Induced Parkinsonism: Evidence from Epidemiological and Experimental Studies"

_biomolecules, 2023, doi:10.3390/biom13081190_

Round 1

Reviewer 1 Report

This is a very timely review highlighting the changing nature of Mn exposures and, correspondingly, the type of neuropathology associated with such exposures.  Review is succinct, yet well-structured. 

One suggestion is to consider including a paragraph identifying areas or research where data are very limited and emerging concerns that are we should be prepared for due to the changing environment and its effects on Mn exposure landscape.

Reviewer 2 Report

The paper titled "Manganese-induced Parkinsonism: evidence from Epidemiological and experimental studies" investigates the association between manganese exposure and Parkinsonism. While the paper contributes to the understanding of this relationship, there are a few points that could be improved or clarified:

·         The introduction should provide a more comprehensive overview of the current understanding of manganese-induced parkinsonism. Discussing the epidemiology, clinical features, and underlying mechanisms would. Discuss the previous studies and their findings regarding the link between chronic manganese exposure and Parkinsonism.

·         The methodology section should provide more details on the search strategy and inclusion criteria for the studies reviewed. Describing the databases searched, the keywords used, and any specific inclusion or exclusion criteria applied would enhance the transparency and reproducibility of the literature review.

·         The paper could benefit from a more detailed description of the statistical analysis conducted. Providing information on the statistical tests employed, adjustments made for confounding factors, and the criteria used for determining statistical significance would help readers understand the strength of the associations reported.

·         The paper could benefit from a more systematic approach to analyzing the shared and distinguishable features. Providing a clear framework or classification system for organizing and comparing the features would help readers understand the basis for the conclusions drawn.

·         Add more description of the environmental and experimental evidence presented. Providing specific details on the environmental assessments conducted, including measurements of manganese levels in the air, water, or other relevant sources, would strengthen the evidence for chronic exposure.

·         The paper could benefit from more detailed descriptions of the diagnostic approaches and their respective strengths and limitations. Providing specific examples of the clinical, radiological, and laboratory features that aid in the diagnosis of manganese-induced parkinsonism would help readers understand the diagnostic process. Please provide a more comprehensive background on manganese-induced parkinsonism, including its epidemiology, clinical features, and the underlying mechanisms of manganese neurotoxicity. This would help readers understand the significance of the diagnosis and the importance of accurate diagnostic criteria.

·         The paper could benefit from more detailed descriptions of the neuropathological findings associated with manganese-induced parkinsonism. Providing specific details on the observed alterations in neuronal structures, cellular pathology, and the distribution of pathology in different brain regions would help readers better understand the neuropathological features.

·         The discussion section should further explore the challenges and potential pitfalls in diagnosing manganese-induced parkinsonism. Discussing the differential diagnosis, including other Parkinsonian syndromes and other neurodegenerative disorders, and highlighting the distinguishing features would provide a more comprehensive analysis.

·         The discussion section should further explore the implications of the neuropathological findings for our understanding of manganese-induced parkinsonism. Discussing the possible mechanisms by which manganese exposure leads to the observed pathology, and how these findings relate to other forms of parkinsonism, would provide a more comprehensive analysis of the topic. It would be beneficial to highlight any unresolved questions or areas for future research in understanding the neuropathology of manganese-induced parkinsonism.

·         The discussion section should further explore the implications of the shared and distinguishable features for the diagnosis, treatment, and prevention of manganese-induced Parkinsonism. Discussing the potential clinical significance and the need for further research in these areas would provide a more comprehensive analysis. Discussing the strengths and limitations of the study, as well as the potential mechanisms linking chronic manganese exposure to Parkinsonism, would provide a more comprehensive analysis and interpretation of the results.

·         Discussing the biological plausibility, including the impact of manganese on dopaminergic pathways and neuroinflammation, would provide a more comprehensive analysis of the findings.

·         The conclusion should summarize the main findings in the tables and figures and provide clear recommendations or implications for clinical practice. It would be beneficial to discuss the importance of early and accurate diagnosis, potential therapeutic interventions, and the need for further research in this area.

·         The conclusion should summarize the main findings and their implications. It would be beneficial to highlight any gaps in the current knowledge and suggest directions for future research, such as the need for longitudinal studies or investigations into specific biomarkers.

Overall, the paper provides valuable insights into the link between chronic manganese exposure and Parkinsonism. Addressing the suggestions and clarifying certain aspects would enhance the understanding and impact of the study. Additionally, discussing the previous literature, providing more details on the environmental and experimental evidence, and considering recent advancements in the field would further contribute to the topic. Moreover, discussing the mechanisms, implications, and recent advancements in the field would further contribute to the topic.
